

# Exploring urban heat islands with a simple thermodynamic model

Mijeong Jeon[1], Kyeongjoo Park[2], Woosok Moon[1], Jae-Jin Kim[1], and Jong-Jin Baik[2]

[1]Division of Earth Environmental System Science, Pukyong National University, Major of Environmental Atmospheric Sciences, Busan, South Korea

[2]School of Earth and Environmental Sciences, Seoul National University, Seoul, South Korea

**Correspondence:** Woosok Moon (woosok.moon@gmail.com)

**Abstract.** The urban heat island (UHI), where urban areas experience higher near-surface temperatures than surrounding rural areas, is becoming a more serious issue in urban climatology due to global warming and rapid urbanization. This study investigates the key mechanisms of the UHI through a simple theoretical thermodynamic model. Using a simple day-night model based on the surface energy balance, we demonstrate that the UHI primarily results from two mechanisms: reduced

diurnal temperature range (DTR) due to larger heat capacity of urban materials and increased mean temperature due to lower urban albedo. These mechanisms explain the reason why the UHI intensity is stronger at night than during the day. The UHI intensity obtained from the theoretical model shows a qualitatively similar diurnal variation to that found in observations, implying the applicability of the theoretical model on understanding the UHI. An analysis of temporal dynamics of UHIs in a megacity (Seoul) and a major city (Suwon) in South Korea shows that the long-term changes in the UHI in both cities are

significantly correlated with those in the urban-rural difference in DTR, highlighting the role of urban heat storage in the UHI. This research provides a theoretical frame for understanding the UHI and its changes with urban characteristics.

## 1 Introduction

Urban areas have higher near-surface temperatures than nearby rural areas (Luber and McGeehin, 2008; Zhao et al., 2014; Chapman et al., 2017). This phenomenon, called the urban heat island (UHI), has become a serious issue in urban environments,

especially with global warming. Global urbanization is expected to continue over the next 30 years, the percentage of urban population being projected to increase from 56% in 2021 to 68% by 2050 (UN-Habitat, 2022). Due to the rapid urbanization, the UHI will likely become more severe, making it an issue that cannot be ignored.

There are many different causes of the UHI. One of the main causes is that impervious materials like asphalt and concrete, used for roads and buildings, absorb and store more heat during the day and release it at night, which raises the temperature

more than nearby rural areas (Wanphen and Nagano, 2009; Ryu and Baik, 2012; Santamouris, 2015; Mohajerani et al., 2017). In addition, the evapotranspiration in urban areas is less than that in rural areas due to limited green spaces and water bodies (Gunawardena et al., 2017; Qiu et al., 2017). This plays a major role in accelerating heat accumulation (Oke, 1982). Moreover, anthropogenic heat resulting from vehicles, industrial activities, and building heating and cooling systems, increases the temperature difference between cities and surrounding areas (Taha, 1997; Grimmond, 2007; Wen and Lian, 2009; Sailor, 2011;

Wang et al., 2023). Air pollutants also contribute to heat accumulation (Grimm et al., 2008; Zhao et al., 2014; Cao et al.,





2016; Han et al., 2020), and tall and densely packed buildings block airflow, further intensifying the UHI (Memon et al., 2008; Rajagopalan et al., 2014; Yuan et al., 2020).

The primary causes of the UHI vary depending on climatic and environmental factors. For example, in regions with hot and dry climates, the lack of evaporative cooling due to low evapotranspiration leads to the development of the UHI (Corumluoglu and Asri, 2015). In humid regions, water vapor in the atmosphere suppresses radiative cooling and slows down the release of heat into the air. In regions with low wind speeds, the stagnation of airflow causes heat to accumulate (Rajagopalan et al., 2014; Mohammed et al., 2020; Al-Obaidi et al., 2021).

Although the UHI intensity differs depending on climatic and environmental factors, there is a common feature: in most regions, the UHI intensity is stronger at night than during the day (Goward, 1981; Stanhill and Kalma, 1995; Fujibe, 1997; Montávez et al., 2000; Runnalls and Oke, 2000; Christen and Vogt, 2004; Nichol et al., 2009; Lee and Baik, 2010; Elagib, 2011; Pal et al., 2012; Berwal et al., 2016; Parker, 2021). In several regions where the UHI is stronger during the day than at night (Ferreira et al., 2013; Borbora and Das, 2014; Mathew et al., 2018), monsoon seasons or frequent heavy rainfall events which bring abundant atmospheric moisture typically appear. In addition to the general feature that the UHI tends to be stronger at night, another universal cause of the UHI across regions is that impervious urban materials increase heat storage. These urban materials accumulate more heat during the day and release it at night, which contributes to the stronger UHI at night (Grimmond and Oke, 1999; Li et al., 2024). This heat storage plays a crucial role in explaining the temporal characteristics of the UHI, the surface energy balance (SEB) needing to be analyzed.

Many previous studies have used the SEB framework to understand the UHI. Oke (1982) introduced causes of the UHI on the basis of the SEB. This study laid the foundation for UHI research. The concept of the urban SEB was well established (Oke, 1988). Grimmond and Oke (1999) emphasized that heat storage is a key factor in strengthening the UHI and quantitatively analyzed how heat is stored in urban environments. Later, Sailor (2011) argued that anthropogenic heat plays a significant role in urban energy balance and stated that anthropogenic heat from transportation and industrial activities strengthens heat accumulation in cities. Recent studies showed that background climate and urban population are significantly associated with the UHI intensity, confirming that background climate and city size are major variables in determining urban SEB (Zhao et al., 2014; Manoli et al., 2019). Additionally, based on global climate model simulations and a SEB theory, Li et al. (2024) revealed that urban areas exhibit stronger temperature persistence compared with rural areas due to impervious materials with large thermal inertia.

The UHI has mainly been explained by focusing on the differences in flux components of the SEB between urban and rural areas. However, while the differences in the magnitude of the SEB terms such as incoming longwave radiation, sensible heat flux, and anthropogenic heat flux between urban and rural areas tend to peak during the day (Oke, 1982, 1988; Li et al., 2015), the UHI is stronger at night. This suggests that the analysis of the SEB based solely on the urban-rural differences in flux terms is insufficient to explain how heat accumulated during the day affects temperatures at night. To address this, it is necessary to analyze the impacts of the SEB on the temporal evolution of temperature by integrating the SEB equation over time, motivating the present study.





Integrating the SEB over time to construct the relationship between the temperature and the heat fluxes is challenging due to the complexity of the SEB. A simplified version of the SEB should be introduced to represent the effect of the heat storage on the temperature. We introduce a day-night model, a simple theoretical thermodynamic model, based on the two-season model introduced by Thorndike (1992) and Moon and Wettlaufer (2012). The two-season model is a simplified model that divides the year into winter and summer to study the seasonal variability of Arctic sea ice. In Thorndike (1992), the two-season model

was first introduced to show the existence of bifurcation in sea ice thermodynamics under global warming. This model was useful for intuitively understanding the dynamics of sea ice growth in relation to atmospheric radiation and climatic forcing. The follow-up study by Moon and Wettlaufer (2012) further investigated why sea ice stabilizes and does not bifurcate when it disappears during the summer, something that Thorndike (1992) could not explain. Although the two-season model used in both studies is a crude approximation, it effectively represents the main seasonal cycle of sea ice. This is because the representative

characteristics found in a complex model can also be extracted using a simpler model. In the Arctic, the summer experiences continuous daylight (polar day) and the winter continuous night (polar night), creating a large cycle based on the presence or absence of solar radiation. This is very similar to the daily cycle, where solar radiation is present during the day and absent at night. Therefore, we expected that by introducing such a simple structure, the mechanism of the UHI could be easily found.

      This study employs a day-night model that divides a day into daytime and nighttime, assuming a constant solar energy

input of $S_0$ during the day and no solar energy input at night. This approach simplifies the complex SEB integration process, allowing for constructing analytic solutions for the daily cycle and diurnal temperature range (DTR) and clearly illustrating how heat accumulation during the day and heat release at night affect the UHI. A control case is then assumed to represent rural conditions. Subsequently, the effects of larger heat capacity of urban materials and lower urban albedo are incorporated into the model to analyze the factors causing and influencing the UHI. These are expected to reveal major mechanisms of the

UHI, explaining why the UHI is more pronounced at night.

      In addition, we utilize observational data in a megacity (Seoul) and a major city (Suwon) in South Korea to compare the diurnal variation of the UHI intensity obtained from the theoretical model and that found in observations and to analyze the relationship between the UHI and urbanization.

      In Sect. 2, we present a simple theoretical thermodynamic model to investigate the mechanisms of the UHI. Section 3

explains methods for obtaining the analytical solutions of the theoretical model and provides details on the observational data used in this study. Section 4 presents the analysis results which include the key mechanisms of the UHI found from the theoretical model and how the UHI intensities in the two cities have evolved with urbanization. Finally, Sect. 5 concludes the present study with a summary of the findings and a discussion of the study's limitations.

## 2   Simple theoretical thermodynamic model

The SEB explains the exchange of energy between the Earth's surface and the atmosphere in a specific region. If the surface energy fluxes are not in balance, the surface temperature changes. Here, the surface temperature means the temperature of a





thin layer including surface and subsurface. Different SEB characteristics and surface temperature between urban and rural areas make weather and climate in urban areas distinct from those in rural areas, resulting in the UHI.

For a thin layer including surface and subsurface, the SEB equation is given by:

$$C\frac{dT_s}{dt} = S(t)(1-\alpha) + L_n - H - LE - G, \tag{1}$$

where $C$ is the heat capacity of the unit surface area of the layer, $T_s$ is the surface temperature, $S(t)$ is the incoming shortwave radiation, $\alpha$ is the surface albedo, $L_n$ is the net longwave radiation, $H$ is the sensible heat flux, $LE$ is the latent heat flux, and $G$ is the ground heat flux (Zeng et al., 2017; Manoli et al., 2019).

The first term on the right-hand side $S(t)(1-\alpha)$ represents the net shortwave radiation at the surface. Urbanization tends to decrease $\alpha$ due to darker surfaces, reduced snow cover in winter, and surface geometry where buildings lead to multiple reflections of incoming shortwave radiation and increase heat storage capacity (Taha, 1997; Lamptey et al., 2005; Yang and Li, 2015). The value of $\alpha$ varies depending on local characteristics. Urban areas generally have lower $\alpha$ compared with rural areas (Oke, 1988; Liu and Wang, 2007; Sugawara and Takamura, 2014), indicating that the net shortwave radiation is generally greater in urban areas compared with rural areas.

The second term $L_n$ consists of downward longwave radiation from the atmosphere to the surface $L_{\downarrow} = \epsilon_a \sigma T_a^4$ and upward longwave radiation from the surface to the atmosphere $L_{\uparrow} = (1-\epsilon_s)\epsilon_a \sigma T_a^4 + \epsilon_s \sigma T_s^4$. Thus, $L_n$ can be written as:

$$L_n = L_{\downarrow} - L_{\uparrow} = \epsilon_s \sigma(\epsilon_a T_a^4 - T_s^4), \tag{2}$$

where $\epsilon_a$ and $\epsilon_s$ are the emissivities of the atmosphere and surface, respectively, $\sigma$ is the Stefan-Boltzmann constant, and $T_a$ is the near-surface air temperature.

The sensible heat flux $H$ is given by $\rho c_p \frac{T_s - T_a}{r_a}$ where $\rho$ is the mean air density, $c_p$ is the specific heat of dry air at constant pressure, and $r_a$ is the aerodynamic resistance (Zhao et al., 2014). It represents the transfer of heat between the surface and the atmosphere due to the temperature difference between them. The latent heat flux $LE$ is expressed as $\lambda \beta g_c \rho \frac{q_{sat,s} - q_a}{r_a + r_c}$ where $\lambda$ is the latent heat of vaporization, $\beta$ is the water stress factor, $g_c$ is the fraction of vegetated area, $q_{sat,s} = q_{sat}(T_s)$ is the specific humidity at saturation, $q_a$ is the specific humidity of air, and $r_c$ is the vegetation canopy resistance (Manoli et al., 2019). It represents the transfer of latent heat between the surface and the atmosphere by evapotranspiration and condensation. $G$ is the ground heat flux between the layer and a subsurface layer below and can be simply expressed as $k\frac{T_s - T_g}{\Delta z}$ where $k$ is the thermal conductivity, $T_g$ is the temperature of the subsurface layer below, and $\Delta z$ is the length between the mid-points of the two layers.

Then, the SEB equation can be expressed as:

$$C\frac{dT_s}{dt} = S(t)(1-\alpha) + \epsilon_s \sigma(\epsilon_a T_a^4 - T_s^4) - \rho c_p \frac{T_s - T_a}{r_a} - \lambda \beta g_c \rho \frac{q_{sat,s} - q_a}{r_a + r_c} - k\frac{T_s - T_g}{\Delta z}. \tag{3}$$

Assuming that the net longwave radiation, sensible heat flux, latent heat flux and ground heat flux are functions of $T_s$ and letting $R \equiv -L_n(T_s, \cdot) + H(T_s, \cdot) + LE(T_s, \cdot) + G(T_s, \cdot)$ simplify the equation to:

$$C\frac{dT_s}{dt} = S(t)(1-\alpha) - R(T_s, \cdot). \tag{4}$$



To understand the mechanisms of the UHI in terms of the SEB, we make the assumption that $T_a = T_s - c$ where $c$ is
a constant. This approach considers that the near-surface (canopy layer) air temperature is primarily driven by the surface
temperature. However, this simplification has the following limitation. According to actual observations, while the canopy
layer UHI is prominent at night, the surface UHI tends to be stronger during the day, making it difficult to assume that $T_a$ and
$T_s$ fluctuate in the same manner. Although temperature change mechanisms can be different between daytime and nighttime
and between the canopy layer and the surface, under the primary objective of understanding the UHI mechanisms in terms of
the SEB holistically, this study utilizes the simplicity and ease of interpretation provided by the assumption. This makes the
model structure simple, the model remaining sufficiently useful for evaluating the quantitative impacts of key factors (e.g., heat
capacity, albedo).

Now, we introduce a day-night model that divides a day into daytime and nighttime with the same duration (12 h) and
assumes a constant solar radiation of $S_0$ during daytime $(0 \leq t < \frac{1}{2})$ and a constant solar radiation of 0 during nighttime
$(\frac{1}{2} \leq t < 1)$. This model is designed to simplify the analysis of energy flux variations over a diurnal cycle focusing on capturing
differences in incoming shortwave radiation between daytime and nighttime. In other words, this model aims to eliminate
complex variabilities and clearly identify essential characteristics of heat accumulation and release processes. This makes it
possible to analytically derive the diurnal variation of the UHI.

The introduction of the day-night model is inspired by the two-season model in the field of sea ice by Thorndike (1992)
and Moon and Wettlaufer (2012). The two-season model divides the year into two seasons, summer and winter, characterized
by two distinct constant values of shortwave radiance. Even with the crude simplification of the seasonal cycle of shortwave
radiance, the periodic solutions contain major features of the seasonal cycle of sea ice thickness. Based on the two-season
model, Thorndike (1992) revealed the existence of a tipping point in sea ice under global warming. Moon and Wettlaufer
(2012) extended the model to show a new feature of sea ice stability in summer. Despite its simplicity, this model has the
advantage of being able to easily extract representative characteristics of the seasonal cycle of sea ice. In this model, total
period is divided into two distinct periods (summer and winter in the two-season model and daytime and nighttime in the
day-night model) and representative values for each period are used, a periodic non-autonomous system being transformed
into two autonomous system. The seasonal cycle in the Arctic, with its midnight sun and polar night, is very similar to the daily
cycle of day and night. Therefore, it is expected that the day-night model can simply extract representative characteristics of
the daily cycle of the UHI. The day-night model can be represented as follows:

$$S(t) = \begin{cases} S_0 & (0 \leq t < \frac{1}{2}) \\ 0 & (\frac{1}{2} \leq t < 1) \end{cases}, \tag{5}$$

where $t = 1$ corresponds to 24 h and the average solar radiation $\bar{S}$ becomes $\frac{1}{2}S_0$.

The simple model proposed in this study is expected to effectively capture key physical mechanisms while explaining the
UHI in a simplified manner. This model has the potential to serve as a useful tool for understanding the UHI and analyzing
characteristics of the UHI under various urban and climatic conditions.




## 3 Methods

### 3.1 Theoretical modeling

In this subsection, the solution of the theoretical model introduced in Sect. 2 is derived. In Eq. (4), $T_s$ is divided into $\bar{T}_s$ (average) and $\eta_s$ (deviation) and $S$ is divided into $\bar{S}$ (average) and $s(t)$ (deviation). Since $\bar{T}_s$ and $\bar{S}$ are time-independent, $\bar{T}_s$ and $\bar{S}$ satisfy the following:

$$F(\bar{S},\bar{T}_s,\cdot) = 0. \tag{6}$$

Here, $F(\bar{S},\bar{T}_s,\cdot) \equiv \frac{1}{2}S_0(1-\alpha) - R(\bar{T}_s,\cdot)$. Considering that the scale of $\bar{T}_s$ is around 300 K which is much larger than that of $\eta_s$ around 10 K, it is possible to apply the Taylor expansion around $\bar{T}_s$ to the Eq. (4), which results in:

$$F(S(t),T_s,\cdot) = F(\bar{S}+s(t),\bar{T}_s+\eta_s,\cdot) \simeq F(\bar{S},\bar{T}_s,\cdot) + s(t)(1-\alpha) + \left.\frac{\partial F}{\partial T_s}\right|_{\bar{T}_s}\eta_s, \tag{7}$$

where $s(t) = \frac{1}{2}S_0\ (0 \leq t < \frac{1}{2})$ or $-\frac{1}{2}S_0\ (\frac{1}{2} \leq t < 1)$. Then, we can rearrange Eq. (4) as:

$$C\frac{d(\bar{T}_s+\eta_s)}{dt} \simeq \left(\bar{S}+s(t)\right)(1-\alpha) - R(\bar{T}_s,\cdot) - \left.\frac{\partial R}{\partial T_s}\right|_{\bar{T}_s}\eta_s. \tag{8}$$

Using Eq. (6):

$$\frac{d\eta_s}{dt} = -\lambda\eta_s + \chi(t), \tag{9}$$

where

$$\lambda = \frac{1}{C}\left.\frac{\partial R}{\partial T_s}\right|_{\bar{T}_s} = \frac{1}{C}\left(-\left.\frac{\partial L_n}{\partial T_s}\right|_{\bar{T}_s} + \left.\frac{\partial H}{\partial T_s}\right|_{\bar{T}_s} + \left.\frac{\partial LE}{\partial T_s}\right|_{\bar{T}_s} + \left.\frac{\partial G}{\partial T_s}\right|_{\bar{T}_s}\right), \tag{10}$$

and

$$\chi(t) = \frac{s(t)(1-\alpha)}{C} = \begin{cases} \frac{S_0(1-\alpha)}{2C} & (0 \leq t < \frac{1}{2}) \equiv \chi_{1/2} \\ -\frac{S_0(1-\alpha)}{2C} & (\frac{1}{2} \leq t < 1) \equiv -\chi_{1/2} \end{cases}. \tag{11}$$

The sensitivity of the temperature-dependent SEB terms is represented by $\lambda$. Here, the sensitivity refers to how these terms respond to small changes in $\bar{T}_s$, which in turn indicates the overall stability of the system. The system stability refers to the ability to return to the original state, which in the context of climate systems means how resilient the system is in response to changing external conditions.

The time-dependent thermal forcing, represented by the daily cycle of the deviation of incoming shortwave radiation, is included in $\chi(t)$. The integration of the Eq. (9) over time leads to:

$$\eta_s(t) = e^{-\lambda t}\int_0^t \chi(t')e^{\lambda t'}dt'. \tag{12}$$



The solution is mainly controlled by the stability $\lambda$. For a bounded solution, it is required that $\lambda > 0$. If not, $\eta_s$ increases exponentially and $T_s$ is unstable to a small perturbation. When $\lambda > 0$, $1/\lambda$ represents a response time-scale of the SEB which can be interpreted as a memory of the system. When the memory is longer, the solar radiation received from sunrise to sunset accumulates for a longer period, allowing the effects of the day to persist into the night. By placing $e^{-\lambda t}$ on the right-hand side of Eq. (12) inside the integral, $\eta_s(t)$ can be expressed as $\int_0^t \chi(t')e^{-\lambda(t-t')}dt'$. Let's consider an example where the current time is 20:00, the sun rises at 06:00, and it set at 18:00. We examine the accumulation process from 06:00 to 20:00. Between 06:00 and 20:00, the forcing of solar radiation, denoted as $\chi(t')$, is applied at each moment. This $\chi(t')$ accumulates in proportion to $e^{-\lambda(t-t')}$. The impacts of $\chi(t')$ at 06:00 and at 18:00 on 20:00 are $e^{-\lambda(20-6)}$ and $e^{-\lambda(20-18)}$, respectively. Normally, the effect of 18:00 is greater than that of 06:00. The larger $\lambda$ is, the smaller the impact, so the effect near sunrise almost disappears with a larger $\lambda$, leaving only the influence near sunset. This corresponds to rural areas that quickly respond to changes in solar radiation. However, the smaller $\lambda$ is, the greater the impact, meaning that the effect during the day can persist longer even after sunset. This implies that solar radiation does not easily escape, which corresponds to urban areas. Here, $\lambda$ is inversely proportional to heat capacity $C$. As the heat capacity $C$ increases due to urban materials in cities, $\lambda$ decreases, leading to an extended memory and the appearance of an accumulation effect.

From past to present, as $t$ becomes sufficiently large, the influence of the initial conditions diminishes and the integral converges to a periodic steady state. Considering that $\eta_S$ is a periodic function, time can be expressed as $t = n + \tilde{t}$. Here, $n$ represents the number of complete cycles (days) and $\tilde{t}$ refers to the time within the current cycle. Therefore,

$$\eta_s(n+\tilde{t}) = \eta_s(\tilde{t}) = e^{-\lambda(n+\tilde{t})}\left[\int_0^n \chi(t')e^{\lambda t'}dt' + \int_n^{n+\tilde{t}} \chi(t')e^{\lambda t'}dt'\right]. \tag{13}$$

The integrals in Eq. (13) are elaborated as:

$$\int_0^n \chi(t')e^{\lambda t'}dt' = \sum_{k=0}^{n-1}e^{\lambda k}\int_0^1 \chi(t')e^{\lambda t'}dt' = \frac{\chi_{1/2}(e^{n\lambda}-1)}{\lambda(e^\lambda-1)}(2e^{\frac{\lambda}{2}}-e^\lambda-1), \tag{14}$$

and

$$\int_n^{n+\tilde{t}} \chi(t')e^{\lambda t'}dt' = e^{\lambda n}\int_0^{\tilde{t}} \chi(t')e^{\lambda t'}dt' = \begin{cases} \frac{\chi_{1/2}}{\lambda}e^{\lambda n}(e^{\lambda\tilde{t}}-1) & (0\leq\tilde{t}<\frac{1}{2}) \\ \frac{\chi_{1/2}}{\lambda}e^{\lambda n}(2e^{\frac{\lambda}{2}}-e^{\lambda\tilde{t}}-1) & (\frac{1}{2}\leq\tilde{t}<1) \end{cases}. \tag{15}$$

The detailed process can be referenced in Moon and Wettlaufer (2013). And the expression for $\eta_s(\tilde{t})$ is represented by:

$$\eta_s(\tilde{t}) = \begin{cases} \frac{\chi_{1/2}}{\lambda}e^{-\lambda\tilde{t}}\left(e^{\lambda\tilde{t}}-1-\tanh\frac{\lambda}{4}\right) & (0\leq\tilde{t}<\frac{1}{2}) \\ \frac{\chi_{1/2}}{\lambda}e^{-\lambda\tilde{t}}\left(-e^{\lambda\tilde{t}}+2e^{\frac{\lambda}{2}}-1-\tanh\frac{\lambda}{4}\right) & (\frac{1}{2}\leq\tilde{t}<1) \end{cases}. \tag{16}$$

The minimum value of $\eta_s$ occurs at $t = \frac{1}{2}$, and the maximum value is observed at $t = \frac{1}{2}$, thus $\eta_{s,min} = -\frac{\chi_{1/2}}{\lambda}\tanh\frac{\lambda}{4}$ and $\eta_{s,max} = \frac{\chi_{1/2}}{\lambda}\tanh\frac{\lambda}{4}$. Therefore, the DTR, defined as $\eta_{s,max} - \eta_{s,min}$, is $2\frac{\chi_{1/2}}{\lambda}\tanh\frac{\lambda}{4}$, implying that the DTR is determined by a nonlinear relationship among the energy absorbed in the surface, the sensitivity of the energy flux balance terms, and the



heat capacity of the surface. The $\tanh x$ function representing the nonlinear relationship has two limiting behaviors. Near the origin, it can be approximated as $y = x$, while it converges to $y = 1$ when $x$ is large enough.

Let's first consider the case when $\lambda$ is sufficiently large. In this case, $\tanh \frac{\lambda}{4} \simeq 1$, so the DTR where $\frac{\lambda}{4} >> 1$ can be expressed as:

$$\text{DTR} = \frac{S_0(1 - \alpha)}{\left. \frac{\partial R}{\partial T_{\rm s}} \right|_{\bar{T}_{\rm s}}} = \frac{2\chi_{1/2}}{\lambda}. \tag{17}$$

This is obtained by the balance between the negative feedback $-\lambda\eta_{\rm s}$ and the thermal forcing $\chi_{1/2}(t)$. The heat reservoir effect measured by the heat capacity $C$ is negligible, thus the deficit or surplus of heat from the SEB is compensated or consumed by the strong negative feedback immediately. Considering various daily cycles of surface temperature in cities, this limit seems to be unrealistic.

Conversely, when $\frac{\lambda}{4}$ is appropriately small, the $\tanh \frac{\lambda}{4}$ function resembles the $y = \frac{x}{4}$ function. Therefore, the DTR can be expressed as:

$$\text{DTR} \simeq 2\frac{\chi_{1/2}}{\lambda}\frac{\lambda}{4} = \frac{\chi_{1/2}}{2} = \frac{S_0(1 - \alpha)}{4C}. \tag{18}$$

In this limit, the negative feedback is much smaller than the heat reservoir term. Thus, some of the incoming shortwave radiation is stored during the day and the stored heat is released to the atmosphere at night, which becomes the major process affecting the DTR. Hence, the DTR is associated with how much heat is stored by materials at the surface. Therefore, as the heat capacity $C$ increases (decreases), the DTR decreases (increases). Similarly, as the albedo $\alpha$ increases (decreases), the DTR decreases (increases). The higher heat capacity in urban areas acts to decrease the DTR, while the lower albedo contributes to increasing the DTR. The contrasting effects of these two factors are quantitatively examined in Sect. 4.1 through an analysis of their relative contributions.

## 3.2 Observational analysis

To compare the diurnal variation of the UHI intensity obtained from the theoretical model and that found in observations and analyze changes in UHI intensity with urbanization, the UHI intensities in Seoul and Suwon are calculated. The Korea Meteorological Administration operates 103 weather observation stations equipped with Automated Synoptic Observing Systems (ASOS) to measure 16 meteorological elements including temperature, wind, relative humidity, and solar radiation. These observations began in 1904, with variations in the measurement period and elements recorded by each station. Since 1972, the majority of stations have consistently collected data. The data are collected at intervals ranging from 4 h to 1 h, with more recent observations tending towards 1-h resolutions. The locations of the 56 weather stations that have continuously collected data from 1972 to the present are shown in Fig. 1.

Prior to data analysis, data preprocessing is conducted to fill missing data up. For short-term missing data occurring at intervals of several hours, such as 4-h interval data from early observations, cubic spline interpolation is used to convert to hourly data. Different methods are used for longer-term missing data. For example, Seoul, the capital of South Korea, has been





under observation since 1908. However, there is a missing period from November 1950 to November 1951 due to the Korean War. This is replaced with the average values from the corresponding dates in 1949 and 1952. In this study, the analysis is conducted after applying these preprocessing steps to all weather stations.

The UHI intensity in Seoul is calculated as the difference in near-surface air temperature between the Seoul and Yangpyeong stations (Kim and Baik, 2004), and the UHI intensity in Suwon is calculated as that between the Suwon and Icheon stations. As shown in Fig. 2, the urban stations are mainly surrounded by built-up areas while the rural stations are mainly surrounded by vegetation and relatively sparsely built-up areas. The distance between the Seoul and Yangpyeong (Suwon and Icheon) stations is 47.5 (44.4) km, and the difference in elevation between them is 38.4 (40.3) m.

South Korea is located in the mid-latitudes of the Northern Hemisphere, with an annual average temperature of around 13°C and a DTR of approximately 8°C (Fig. 3). In South Korea, distinct seasonal characteristics and large temperature variations appear in a year, with hot and humid summers, cold and dry winters, and mild temperatures in spring and autumn. The territory encompasses diverse topography including mountainous regions, coastal areas, and plains. Coastal regions experience relatively smaller temperature fluctuations due to the influence of maritime climate.

Since the 1970s, South Korea has undergone explosive urbanization and industrialization due to economic development. Among these changes, Seoul, which currently houses approximately 18.3% of the country's total population, has emerged as a center of development while experiencing particularly rapid urbanization. In Seoul's case, a sharp rise in temperature could be observed during the rapid development period of the 1980s (Fig. 4), indicating that urbanization has a significant impact on the long-term increasing trend of temperature.

## 4 Results

### 4.1 Mechanisms of the UHI

To explore the mechanisms of the UHI using the theoretical model, the diurnal variations of urban and rural near-surface air temperatures are examined. For calculations with the theoretical model, the values of $S_0$, $C$, $\alpha$, $\epsilon_s$, $c$, and $\lambda$ are needed.

$S_0$ is set to 900 W m$^{-2}$, $\epsilon_s$ is set to 0.95, and $c$ is set to 6.5 K for both urban and rural areas. To consider the effects of larger urban heat capacity and lower urban albedo on the UHI, $C$ and $\alpha$ are set to 21 (14) J m$^{-2}$ K$^{-1}$ and 0.15 (0.16) for the urban (rural) area, respectively. $\lambda$ is dependent on $\left.\frac{\partial R}{\partial T_s}\right|_{\bar{T}_s}$, which represents the background climate. To obtain representative values of $\left.\frac{\partial R}{\partial T_s}\right|_{\bar{T}_s}$ for typical urban and rural areas, a two-dimensional idealized numerical simulation is performed using the Weather Research and Forecasting (WRF) model (Skamarock et al., 2019) coupled with the Seoul National University Urban Canopy Model (SNUUCM) (Ryu et al., 2011). The simulation set-up is the same as that adopted in Park et al. (2024) except that the simulation period is 0000 LST 19 March–0000 LST 21 March (24-h spin-up period) and the initial surface potential temperature is set to 288.15 K. From this numerical simulation, the values of $\left.\frac{\partial R}{\partial T_s}\right|_{\bar{T}_s}$ are obtained 11.8 and 30.9 W m$^{-2}$ K$^{-1}$

for the urban and rural areas, respectively. Thus, we first conducted calculations where $\left.\frac{\partial R}{\partial T_s}\right|_{\bar{T}_s}$ is set to 30.9 W m$^{-2}$ K$^{-1}$ for



both urban and rural areas to isolate the effects of $C$ and $\alpha$. Next, to account for the background state differences between urban and rural areas, we additionally conduct calculations where $\left.\frac{\partial R}{\partial T_s}\right|_{\bar{T}_s}$ is set to 11.8 (30.9) W m$^{-2}$ K$^{-1}$ for urban (rural) areas. In the first case, the value of $\lambda$ used in the theoretical model is 1.5 (2.2) s$^{-1}$ for urban (rural) areas, while in the second case, it is 0.6 (2.2) s$^{-1}$. This implies that in both cases, the system stability of urban areas is lower than that of rural areas.

Now, let us compare the effects of heat capacity and albedo on DTR and examine how each factor influences the UHI. In the first case, where the background state is assumed to be the same, varying only $C$ (Fig. 5a) keeps the average temperature similar between urban and rural areas, but the DTR in urban areas is smaller by 3.67°C compared to rural areas. When only $\alpha$ is varied (Fig. 5b), the DTR in urban and rural areas remains nearly the same, while the average temperature in urban areas increases by 0.86°C. Thus, varying both $C$ and $\alpha$ (Fig. 5c) leads to a decrease in DTR and an increase in average temperature.

In the second case which considers background state differences, the overall results remain similar except that the lower alpha in urban areas increases the DTR by 1.18°C compared to rural areas (Fig. 6b). However, since the reduction in DTR due to the increase in $C$ (Fig. 6a) outweighs the increase caused by the decrease in $\alpha$, the final result still shows a decrease in DTR (Fig. 6c).

In conclusion, when urbanization occurs, the increase in heat capacity leads to a reduction in DTR as past effects accumulate over longer periods. Here, the temperature in urban areas is higher than in rural areas at night, but during the day, the temperature in urban areas can be lower than that in rural areas. This phenomenon occurs because the high heat capacity of urban surfaces allows them to store a significant amount of heat and to be warmed more slowly than rural areas, leading to the urban cool island (UCI) during the day. The causes of the UCI have been similarly discussed in Giovannini et al. (2011), Ganbat et al. (2013), and Santamouris (2015). Additionally, the decrease in albedo does contribute to an increase in DTR, but its effect is minimal. Therefore, among the increase in heat capacity and the decrease in albedo, the increase in heat capacity has a greater influence on DTR, while the decrease in albedo plays a more significant role in increasing the average temperature. When these two changes co-occur, there is little temperature difference between urban and rural areas during the day but the temperature difference is pronounced at night, meaning that the UHI is stronger at night (Fig. 5c, 6c).

Furthermore, urban vulnerability is closely linked to decreased system stability, suggesting that cities will be more vulnerable when global warming occurs. Vulnerability is defined by the magnitude of $\lambda$ in Eq. (10), where $\lambda$ is inversely proportional to heat capacity $C$. Therefore, the high heat capacity of cities leads to smaller $\lambda$ values, resulting in reduced system stability, causing temperatures to react more dramatically to even small external factors. Indeed, McCarthy et al. (2010) revealed that cities have high vulnerability to climate change.

This study has limitations in quantitative analysis of various heat fluxes. $\lambda$ is influenced not only by heat capacity but also by various combined fluxes and can significantly vary depending on regional, seasonal, and temporal factors. However, we assume it as a constant value for urban and rural areas, which limits the ability to fully capture the subtle variabilities present in actual urban environments.

Nevertheless, the main purpose of this study is to qualitatively understand the key mechanisms of the UHI by simplifying the problem. To achieve this, we track the core physical processes of the UHI by introducing a simple day-night model to the SEB equation and present a simplified theoretical framework applicable even in complex environments. In particular, we



demonstrat that this approach is also effective in understanding the diurnal variability of the UHI by incorporating the concept
of the two-season model, which was used to explain the seasonal variability of sea ice, into our day-night model. So, the results
of this study not only contribute to a qualitative understanding of the main characteristics of the UHI but also suggest that this
theoretical approach can be utilized in various environmental and climate research fields in the future.

## 4.2  Diurnal variations of UHI intensity

In this subsection, we focus on showcasing the similarity between our theoretical model's predictions and the observed data.
Our findings argue that urbanization can be theoretically articulated and its effects are quantitatively mirrored in observed data.

Seoul is selected as an urban area, while Yangpyeong is selected as rural area. Seoul, as South Korea's representative city,
reflects the characteristics of urbanization. As of 2024, Seoul's population density is 15,506 people/km$^2$, approximately 107
times that of Yangpyeong (144 people/km$^2$). This stark difference in population density provides an important background for
comparing the effects of urbanization between the two regions.

The core of our analysis lies in the qualitative agreement between the model results and the observed UHI in Seoul and
Yangpyeong. The model, grounded in fundamental principles, captures the phenomenon where urban areas retain heat accu-
mulated during the day, resulting in significantly higher nighttime temperatures compared to rural areas (Fig. 7a). This pattern
is consistently reproduced in the model even when background state differences between urban and rural areas are considered
(Fig. 7b). The observational data also show that nighttime temperatures in Seoul are higher than those in Yangpyeong (Fig.
7c). Additionally, in both the model and observational data, a temporary UCI appears at peak daytime temperatures, which is
expected to result from the high heat capacity of urban materials, as shown in Fig. 5a. These results indicate that Seoul has a
significant heat storage effect, which is consistently captured by both the SEB model and the observational data.

The significance of our study lies in showing that the seemingly complex UHI can be fundamentally interpreted and ex-
plained through a simple theoretical model. While the model's monotonic pattern and differences in the timing of peak tem-
peratures highlight the limitations of the simplified day-night model compared to actual observations, the fundamental charac-
teristic of the UHI—where daytime temperatures are similar, but nighttime temperatures in rural areas are significantly lower
than those in urban areas—is consistently captured. This approach not only validates the theoretical foundation of our model
but also enhances its practical relevance.

## 4.3  Long-term changes in UHI inensity in Seoul and Suwon with urbanization

This subsection analyzes changes in the UHI intensity in a megacity (Seoul) and a major city (Suwon). It also compares
the effects of different urbanization rates. Seoul has experienced rapid urbanization since the 1970s due to industrialization
and population growth. Suwon also began urbanizing in the 1970s, but its development progressed more gradually, with key
milestones such as the formation of a central business district in the 1980s and new city developments in the 1990s. Based
on this, the study examines the patterns of the UHI, DTR, and average temperature changes to find the relationship between
urbanization and UHI. To analyze these changes, Yangpyeong is selected as the rural reference area for Seoul, and Icheon is
selected for Suwon. Since they have similar topographical features, making them suitable for comparison.





For Seoul, the late-night to early morning UHI intensity exhibits an increasing trend between 1972 and 1990 ($p < 0.05$; Fig. 8a). This pattern matches the urbanization process in Seoul and aligns with the SEB model's prediction of increased heat storage caused by urbanization (Fig. 8b). On the other hand, during the same period, the afternoon UHI intensity exhibits a
decreasing trend ($p < 0.05$). The SEB model analysis (Fig. 5a) also suggests that an increase in heat capacity in urban areas can cause a temporary UCI during the day. These changes in UHI and UCI correspond to the increasing DTR difference and average temperature difference between Seoul and Yangpyeong (Fig. 8b).

After 1990, the UHI pattern in Seoul began to change. The daytime UHI intensity exhibits an insignificant trend, but the nighttime UHI intensity exhibits a decreasing trend ($p < 0.05$; Fig. 8a). It is also seen that both the DTR difference and the
average temperature difference decline. This may be associated with the overall decreasing urban-rural population difference between Seoul and Yangpyeong which indicates the migration of people to nearby areas and the urbanization of rural areas like Yangpyeong (Fig. 8c). However, explaining UHI changes based only on population growth has limitations. Thus, a further research is needed to address this.

For Suwon, there was little difference in temperature between Suwon and Icheon in 1972. The UHI intensity is weak both
during the day and at night (Fig. 9a). However, from 1972 to 2022, the nighttime UHI intensity exhibits an increasing trend ($p < 0.05$), while the daytime UHI intensity exhibits an insignificant trend. More specifically, from 1972 to 1990, the nighttime UHI intensity exhibits an increasing trend ($p < 0.05$), while the daytime UHI intensity exhibits a decreasing trend ($p < 0.05$). The increasing trend in nighttime UHI intensity becomes more prominent from 1990 to 2000 ($p < 0.05$), whereas the trend in daytime UHI intensity becomes statistically insignificant. Since the 2000s, both daytime and nighttime UHI intensities exhibit
insignificant trends. The increasing trend in UHI intensity is align with those in the DTR difference and average temperature difference between Suwon and Icheon (Fig. 9b), possibly associated with the overall increasing urban-rural population difference (Fig. 9c). These trends match the predictions of the SEB model.

Comparing the UHI changes in Seoul and Suwon shows that the UHI growth and stabilization follow different patterns depending on the stage of urbanization. Seoul exhibits an increasing trend in UHI intensity from 1972 to 1990 but later with
the urban-rural population difference decline, UHI intensity has exhibited a decreasing trend. In contrast, Suwon exhibits an increasing trend in UHI intensity as urbanization has progressed since the 1970s. At a certain level, even if urbanization continues, the UHI growth may slow down or stabilize. If urban development continues in the future, further studies are needed to analyze the saturation point and stabilization mechanisms of the UHI.

## 5  Conclusions

This study demonstrates how urbanization influences the formation of the UHI through a simplified theoretical model. Additionally, it provides a comprehensive analysis of UHI evolution in a megacity (Seoul) and a major city (Suwon), where urban development processes differ significantly. The findings offer critical insights into the mechanisms underlying UHI formation.

First, we construct a theoretical framework that effectively explains the core mechanisms of the UHI by integrating the SEB theory with a simplified day-night model (Fig. 10). The results confirm that the primary drivers of the UHI are the reduction





in DTR due to the increased heat capacity of urban surface materials and the increase in average temperature due to lower albedo in urban areas. The combination of these factors results in a UHI characterized by higher nighttime temperatures in cities compared to rural areas, while also leading to the occurrence of a temporary UCI during the day.

Second, to validate the SEB model against real-world observations, we compare theoretical predictions with observational data from Seoul (urban) and Yangpyeong (rural). Both the model and the data consistently exhibit a strong heat storage effect, 375 indicating that heat absorbed during the day is released at night, intensifying the UHI at night. Additionally, an observed temporary UCI during the day supports the predictions of the simplified SEB model.

To further analyze UHI evolution, we compare two cities with different urbanization trajectories. In Seoul, which has experienced rapid urbanization from the 1970s, the nighttime temperature exhibits an increasing trend while the daytime temperature exhibits a decreasing trend between 1972 and 1990. However, after reaching saturation in the 1990s, the UHI intensity between 380 the urban and rural areas began to decline. While this pattern is matched with population changes, additional factors likely contribute, necessitating further investigation. In contrast, Suwon, which has developed gradually, exhibits a gradual increase in nighttime temperature differences in parallel with population growth and urban expansion. Moreover, the observed reduction in DTR and increase in average temperature align well with SEB model predictions. These findings highlight the significant impact of urban development patterns on UHI evolution.

Incorporating a simplified day-night model into the SEB framework, this study demonstrates the utility of theoretical models in explaining complex UHI and provides a foundation for predictive tools in future research. While this study establishes a theoretical framework for understanding UHI, several challenges remain. The SEB model expresses temperature as a function of surface temperature ($T_s$) and subsequently assumes a linear relationship between surface and air temperatures ($T_a = T_s - c$) for analytical purposes. This assumption represents a clear limitation of the model, as it does not fully capture the complex 390 interactions between surface and atmospheric processes. However, this simplification is necessary to construct a model that maintains computational efficiency while retaining key physical mechanisms. Furthermore, this study has the limitation of not accounting for anthropogenic heat in the calculation process. Incorporating the effects of anthropogenic heat into the theoretical model is expected to help to enhance the understanding of the UHI mechanisms.

Additionally, by utilizing ASOS data from South Korea, this study provides valuable insights into UHI dynamics in mid-395 latitude regions and can contribute to a broader understanding of the UHI by extending research to other areas with similar climatic and urbanization characteristics. Given that UHI patterns vary depending on urban development processes, further research is required to refine the model's applicability to diverse urban environments. Expanding this study to other mid-latitude regions and different urban settings will enhance its relevance and impact.

In conclusion, this study bridges theoretical modeling and empirical observations to provide a comprehensive understanding 400 of the UHI. The findings reinforce the necessity of sustainable urban development and establish a foundation for advancing knowledge on complex urban climate dynamics.

*Code availability.* The code used in this study is available from the corresponding author upon reasonable request.



*Data availability.* The observational data used in this study are publicly available from the Korea Meteorological Administration (KMA) via the Weather Data Open Portal at https://data.kma.go.kr/data/grnd/selectAsosRltmList.do?pgmNo=36.

*Author contributions.* Mijeong Jeon contributed to the original draft writing, software development, data curation, validation, investigation, visualization, and formal analysis. Kyeongjoo Park contributed to software development, validation, investigation, and manuscript editing. Woosok Moon contributed to conceptualization, methodology development, formal analysis, supervision, and manuscript editing. Jae-Jin Kim contributed to supervision and manuscript editing. Jong-Jin Baik contributed to supervision and manuscript editing.

*Competing interests.* The authors declare that they have no conflict of interest.

*Acknowledgements.* This research was supported by the Korea Meteorological Administration Research and Development Program (grant no. RS-2024-00404042) and by the Global Learning & Academic research institution for Master's·PhD students, and Postdocs (LAMP) Program of the National Research Foundation of Korea (NRF) funded by the Ministry of Education (grant no. RS-2023-00301702). This work also benefited from partial support through the 2023 Capstone Design course.



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





**Figure 1.** Locations of 56 Automated Synoptic Observing Systems (ASOSs) used in this study. The urban (Seoul and Suwon) stations and rural (Yangpyeong and Icheon) stations used for analysis of the UHI are indicated by orange and blue dots, respectively. Map lines delineate study areas and do not necessarily depict accepted national boundaries.





(a)

(b)

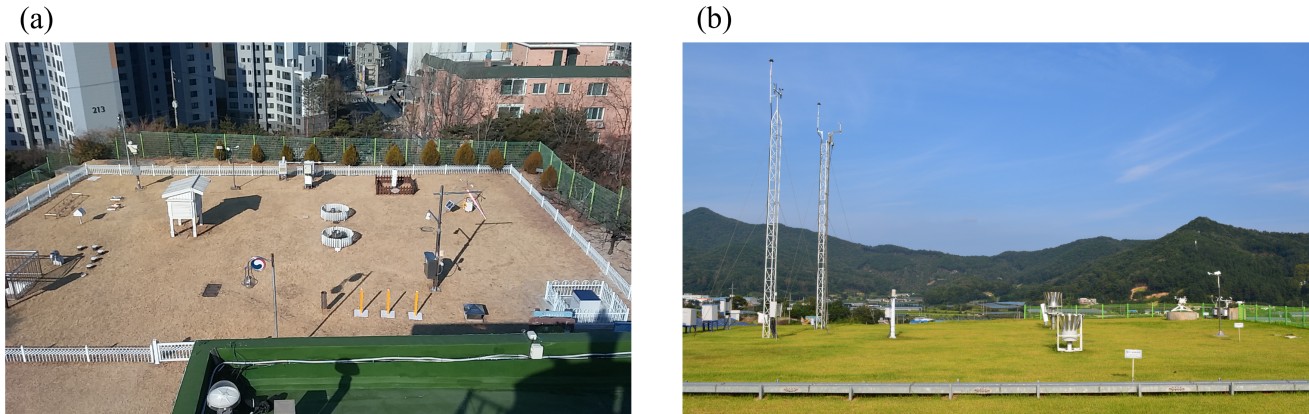

**Figure 2.** Two observation sites in (a) an urban area (Seoul) and (b) a rural area (Yangpyeong). Photographs provided by the Korea Meteorological Administration, Seoul Regional Office of Meteorology.

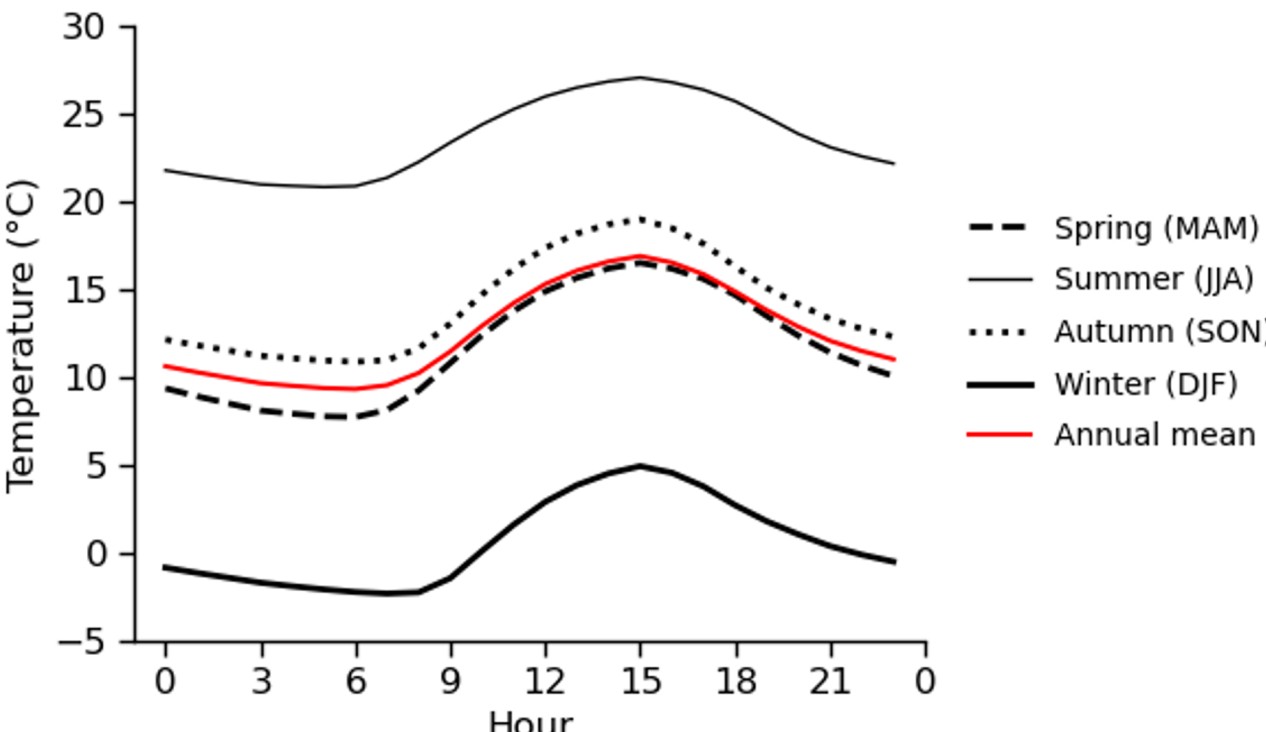

**Figure 3.** Diurnal variation of near-surface air temperature for each season (black) and the annual mean (red), averaged over 51 years (1972–2022) and 56 stations.



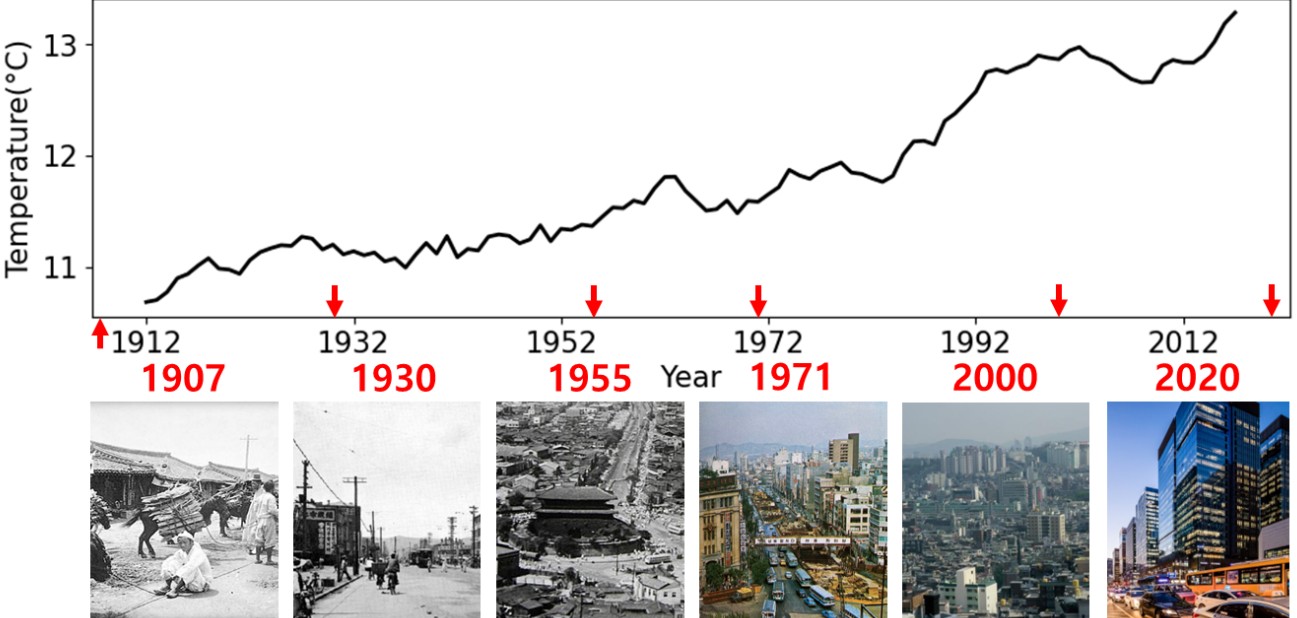

**Figure 4.** Long-term changes in annual average near-surface air temperature in Seoul. The images below show adjacent areas to Seoul weather station in 1907, 1930, 1955, 1971, 2000, and 2020. Photographs for 2000 and 2020 are reproduced with permission from Seoul 2000, 2020 Urban Form and Landscape, Seoul Metropolitan Government. Photographs from 1907, 1930, and 1955 are in the public domain under Korean copyright law (published before 1977).

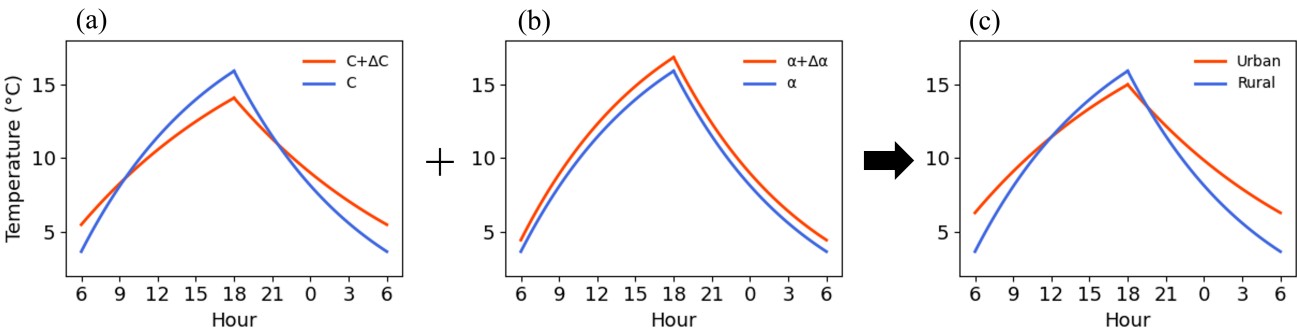

**Figure 5.** (a) Effects of increase in heat capacity on the diurnal variation of the UHI, (b) effects of reduction in albedo on the diurnal variation of the UHI, and (c) effects of both increase in heat capacity and reduction in albedo on the diurnal variation of the UHI found from the theoretical model under the same background state in urban and rural areas.



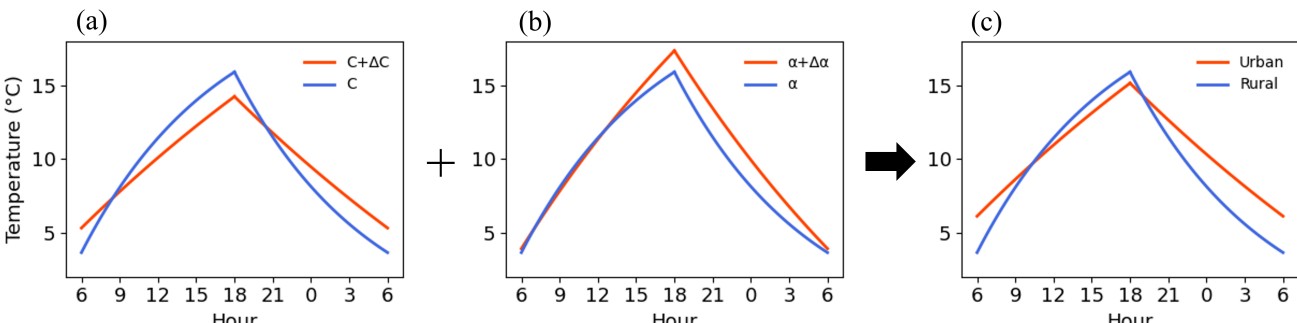

**Figure 6.** Same as Fig. 5 except for different background state in urban and rural areas.

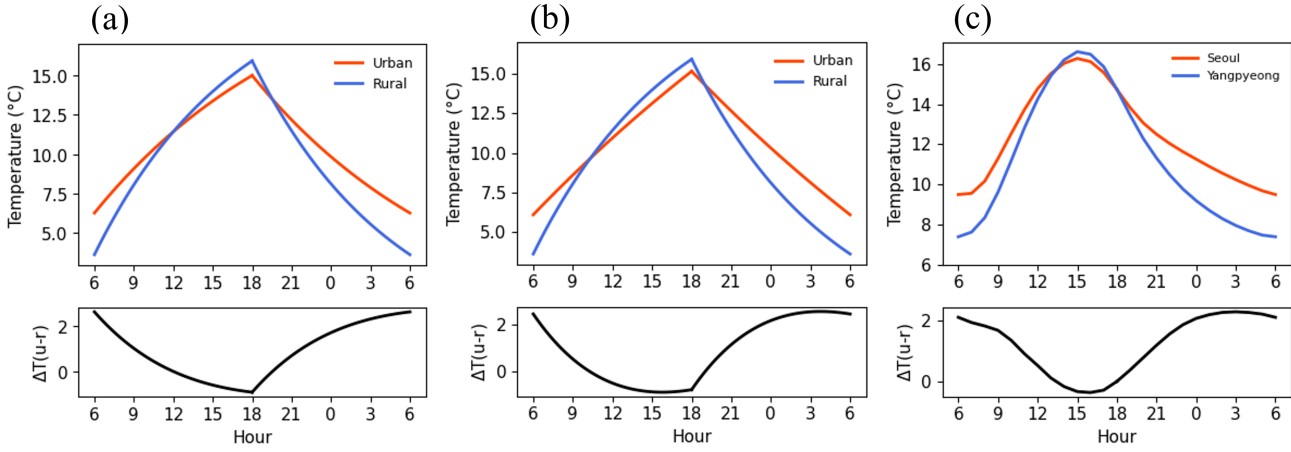

**Figure 7.** (a) Diurnal variations of urban (orange) and rural (blue) near-surface air temperatures and UHI intensity (black) calculated by the theoretical model under the same background state. (b) Same as (a) except for different background state. (c) Same as (a) except for those found in observations at Seoul (orange) and Yangpyeong (blue) stations.





**Figure 8.** (a) Long-term changes in annual average UHI intensity in Seoul (shade) for each hour, (b) long-term changes in annual average UHI intensity in Seoul (pink) and urban-rural difference in DTR (green), and (c) long-term changes in urban-rural difference in population between Seoul and Yangpyeong.





**Figure 9.** Same as Fig. 8 except for Suwon and Icheon.





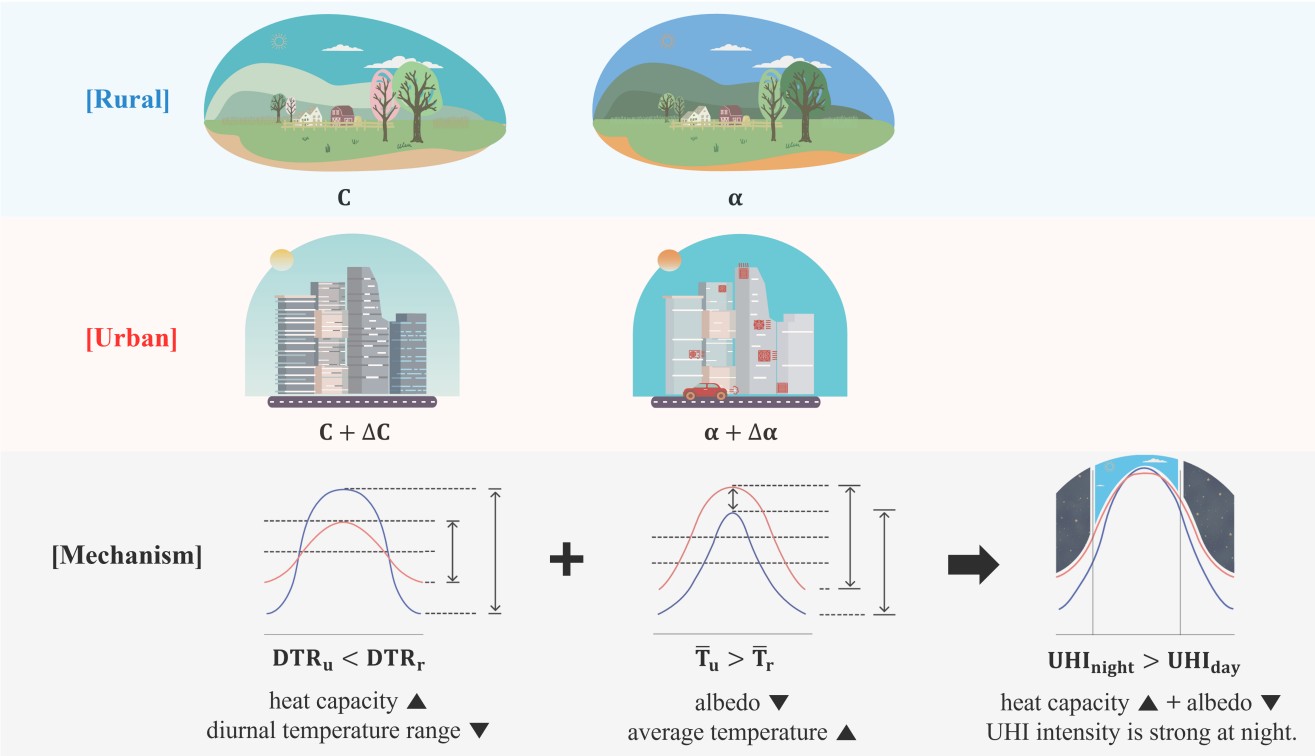

**Figure 10.** Schematic diagram of the main mechanisms of the UHI. In this study, we analyze the mechanisms of the UHI using a SEB model coupled with a simple day-night model. Compared to rural areas, urban areas exhibit larger heat capacity ($\Delta C$) due to man-made materials such as concrete and asphalt. Due to the larger heat capacity in urban areas, heat is more stored during the day and is more released at night, leading to a reduction in DTR. In addition, due to lower albedo in urban areas ($\Delta \alpha$), more energy is absorbed at the surface, raising the average temperature in urban areas. These two processes - "decreased DTR due to larger urban heat capacity ($\Delta C$)" and "increased average temperature due to lower urban albedo ($\Delta \alpha$)" - make the UHI particularly strong at night. The diurnal variation of the UHI obtained from observational data is qualitatively similar to that obtained from the theoretical model.