# Peer review of "Exploring urban heat islands with a simple thermodynamic model"

_EGUsphere, 2025_

## Author Comment (AC1)

**General comments:**

This study constructed a simple theoretical thermodynamic model that explains the key mechanisms of the urban heat island (UHI) by integrating the surface energy balance (SEB) theory with a simplified day-night model. Using this approach, the authors demonstrated that the UHI primarily results from two factors: the reduced diurnal temperature range (DTR) due to the larger heat capacity of urban materials, and the increased mean temperature associated with lower urban albedo. The study further compared the model's theoretical predictions with observed UHI characteristics and long-term changes in a megacity (Seoul) and a major city (Suwon) in South Korea. The results showed that long-term changes in the UHI in both cities are significantly correlated with the urban-rural difference in DTR, highlighting the role of urban heat storage in UHI intensity.

Even today, when sophisticated urban models incorporating complex processes are widely used, I agree that simple models can still be effective, depending on the objective. By omitting minor factors and focusing on dominant processes, such models can distill the essence of the phenomenon and provide important insights. At the same time, however, it is crucial to clearly demonstrate the novelty of the presented results and conclusions to meet the standards of a scholarly article, regardless of the methods employed.

I have carefully read the paper several times and re-examined the related literature to evaluate the novelty of this study and to identify its potential contributions to urban climate and land surface process research. However, the novelty of the study is not clearly demonstrated and could not be identified. The following points, in particular, require improvement.

We sincerely thank for the careful reading of our manuscript and for the thoughtful and constructive comments. We fully understand the concern regarding the novelty of our study. In response, we have thoroughly revised the manuscript to emphasize that our simple and intuitive model provides a clear way to verify well-known mechanisms of the urban heat island (UHI). We also highlight the potential contribution of such an approach, as simple models can be particularly useful for identifying and characterizing representative features across diverse fields. We believe that these revisions clarify the contribution of our study and help to address the reviewer's concern.

**Specific comments:**

[1]

A key issue in evaluating this paper is whether the novelty of introducing a new model to explain already well-known mechanism should be acknowledged. The manuscript proposed a theoretical model based on SEB theory, proceeding from the view that the key factors of the UHI mechanism are the large heat capacity of urban materials and urban albedo (specifically, the lower albedo typically found in cities), and then evaluated the sensitivity of surface air temperature to these two factors. The model produces the following result:

1.369-372

"The results confirm that the primary drivers of the UHI are the reduction in DTR due to the increased heat capacity of urban surface materials and the increase in average temperature due to lower albedo in urban areas. The combination of these factors results in a UHI characterized by higher nighttime temperatures in cities

compared to rural areas, while also leading to the occurrence of a temporary UCI during the day."

However, this description reflects knowledge that is already widely recognized in the field. The importance of heat capacity as a primary driver of the UHI was first emphasized by Mitchell (1961), and then many studies have conducted quantitative assessments of the contribution of heat capacity to the UHI using mathematical models (e.g., Myrup, 1969; Oke, 1982; Oke, 1987; and Grimmond and Oke, 1999). While the simple model of Myrup (1969) failed to reproduce the larger nighttime UHI, Oke (1982) provided a mathematical explanation for this phenomenon. Similarly, the effects of albedo have been quantitatively evaluated in numerous studies (e.g., Sailor, 1995). Atkinson (2003) compared the effects of several factors, including heat capacity and albedo, on the UHI, though his conclusions differ from those presented in this study. The studies introduced here are limited to pioneering studies; numerous investigations have subsequently been carried out employing different models and focusing on various cities (e.g., Adachi et al., 2016). As a result, previous studies have already established the following points reiterated here: (i) heat capacity is a principal driver of the UHI, (ii) nighttime UHI typically exceeds daytime UHI, and (iii) the fundamental effects of heat capacity and albedo on surface air temperature.

The manuscript states that the proposed model aims to provide a qualitative rather than a quantitative understanding. However, as noted above, qualitative aspects are already well understood, while current research largely seeks quantitative insights based on detailed observations and sophisticated models. Against this backdrop, the authors should clearly articulate what advantages this model offers.

- Mitchell, J. M. (1961): The Temperature of Cities. Weatherwise, 14(6), 224–258. https://doi.org/10.1080/00431672.1961.9930028
- Myrup, L. O. (1969): A Numerical Model of the Urban Heat Island. J. Appl. Meteor.
   Climatol., 8, 908-918, https://doi.org/10.1175/1520-0450(1969)008<0908:ANMOTU>2.0.CO;2.
- Oke, T.R. (1982): The energetic basis of the urban heat island. Q.J.R. Meteorol. Soc., 108: 1-24. https://doi.org/10.1002/qj.49710845502
- Oke, T.R. (1987): Boundary Layer Climates. Routledge. pp464. https://doi.org/ 10.4324/9780203407219
- Grimmond, C. S. B., and T. R. Oke, (1999): Heat Storage in Urban Areas: Local-Scale Observations and Evaluation of a Simple Model. J. Appl. Meteor. Climatol., 38, 922–940, https://doi.org/10.1175/1520-0450(1999)038<0922:HSIUAL>2.0.CO;2.
- Sailor, D. J. (1995): Simulated Urban Climate Response to Modifications in Surface Albedo and Vegetative Cover. J. Appl. Meteor. Climatol., 34, 1694–1704, https://doi.org/10.1175/1520-0450-34.7.1694.
- Atkinson, B. (2003): Numerical Modelling of Urban Heat-Island Intensity. Boundary-Layer Meteorology, 109 (3), pp 285-310.
- Adachi, S. A., F. Kimura, H. G. Takahashi, M. Hara, X. Ma, H. Tomita, (2016): Impact
  of high-resolution sea surface temperature and urban data on estimations of surface
  air temperature in a regional climate. J. Geophys. Res., 121(18), 10486-10504,
  doi:10.1002/2016JD024961.

We agree that the mechanisms highlighted in our original description—namely, the reduced diurnal temperature range (DTR) due to the increased heat capacity of urban materials and

the rise in mean temperature due to lower albedo—are already well recognized in the field. We acknowledge that simply reiterating these known mechanisms would not provide sufficient novelty for this study.

In response, we have thoroughly revised the manuscript to more clearly articulate the methodological contributions of our work. Specifically, our approach integrates the surface energy balance (SEB) framework with a simplified day–night model and introduces a time-integrated perspective to evaluate the role of cumulative energy storage in shaping temperature variations. By converting a periodic non-autonomous system (day–night cycle) into two autonomous systems, the model allows the dominant characteristics of the UHI to be isolated and examined in a transparent manner.

This methodological framework provides two key advantages. First, it offers an intuitive and physically grounded way to connect the energy storage process not only with the reduction in DTR but also with the rise in mean temperature, showing that phenomena commonly reproduced by complex urban models can also be explained within a simple theoretical framework. Second, it demonstrates the potential utility of simple models in identifying representative physical processes across diverse research contexts, not only in urban climatology but also in other fields dealing with periodic forcing and cumulative effects.

The introduction of a simple model is not intended to propose new mechanisms or perspectives, but rather to demonstrate that such a framework can encapsulate processes that have already been identified in previous studies. Within this context, the fundamental nature of the urban heat island effect can be interpreted in a more transparent and tractable manner. While the primary causes of the urban heat island may differ among cities, the formalism of this simple model provides a coherent basis for offering deeper and more generalized interpretations across diverse urban environments.

We believe that these revisions better highlight the distinct contribution of our study and clarify its relevance in the broader context of urban climate research.

Related to comment [1], the manuscript requires a stronger review and citation of relevant prior work. Urban climate research has a long history and an extensive body of literature. There are multiple review papers (e.g., Kanda, 2007 and the list in Table 1 of that paper) that could serve as useful starting points. At a minimum, a more thorough survey of key prior studies directly relevant to the objectives of this work is needed. Such a review would help clarify the paper's originality, position its contribution within existing knowledge, and highlight its significance to urban climate research.

 Kanda, M. (2007): Progress in Urban Meteorology :A Review. Journal of the Meteorological Society of Japan, 85B, 363-383. https://doi.org/10.2151/jmsj.85B.363

We agree that our original manuscript did not sufficiently reflect the extensive literature on urban climate research. After reviewing the paper suggested by the reviewer along with several key prior studies, we have revised the Introduction to include a more thorough discussion of relevant works. These additions strengthen the contextual background of our study and help to clarify its originality and significance within the existing body of knowledge. The revisions are reflected in lines 48–64 of the revised manuscript.

In the latter part of the paper, the authors compare the qualitative insights from their model with observations. However, the interpretations rely mainly on fundamental, well-known ideas, such as the reduction in DTR due to the large heat capacity of urban areas and the rough correlation between UHI intensity and population density. Sections 4.1 and 4.2 lack sufficient detail about the experimental setup and justification for parameter values. As a result, there is not enough information to interpret the results or replicate the analysis, which makes it difficult to assess the validity of the findings. Moreover, Section 4.3 is based on a largely speculative discussion of the observational results, without adequate evidence to support its claims.

As the reviewer rightly pointed out, the original manuscript did not provide sufficient explanation regarding the model experimental setup and the rationale for parameter selection. Accordingly, Sections 4.1 and 4.3 have been thoroughly revised to include more detailed descriptions and interpretations.

In Section 4.1, we have clearly described the physical background of the experiment and the process of setting key input parameters. The conditions under which the values of  $\frac{\partial R}{\partial T}|_{\bar{T}_s}$

were derived from the WRF–SNUUCM numerical simulation are now specified. Although these values may vary over time, we clarified that their temporal variability has a negligible impact on the results within the main time scales considered in this simplified model. In addition, the parameters c,  $S_0$ ,  $\epsilon_s$ ,  $\alpha$ , and C were selected to fall within physically reasonable ranges representative of typical urban and natural surfaces. We also emphasized that the objective of this study is not to reproduce exact numerical values, but rather to elucidate the relative magnitude and underlying physical mechanisms between urban and rural conditions. Therefore, moderate variations in parameter values do not substantially affect the key findings or interpretations.

In Section 4.3, we expanded the discussion by adding detailed interpretations of the observational results and broadening the scope of analysis.

In summary, the main focus of this study lies in explaining the relative characteristics and their physical linkages between urban and rural environments within a simplified theoretical framework, rather than achieving precise numerical reproduction. We thus expect that the revised manuscript sufficiently addresses the reviewer's concerns.

**Minor Concerns:**

[1] What is the reason for the sudden change in the UHI of Suwon shown in Fig. 9(a) around 1998? It is worth considering whether the continuity of the observational data may have been affected by factors such as relocation of the observation sites or environmental changes in their immediate vicinity.

We would like to explain the sudden change in the UHI intensity of Suwon around 1998 indicated in Fig. 9(a). It is possible that environmental changes near the Suwon station affected this variation. A large residential complex (apartment buildings) was constructed about 200 m from the station in 1984, and over time this may have significantly changed the urban characteristics of the surrounding area, which could have influenced the long-term increasing trend of the UHI intensity.

We also examined whether the station itself was relocated. According to the Korea Meteorological Administration, the Suwon station was located at the same site (longitude 126.98533, latitude 37.27226, altitude 34.84 m) from January 1, 1964, to July 24, 2019. During this period, no discontinuity in the observational data occurred due to station relocation. Since July 24, 2019, the station has been located at another site about 1.66 km away (longitude 126.98300, latitude 37.25746, altitude 39.81 m) with a similar environment. However, this relocation is not directly related to the main patterns analyzed in this study, and the environmental difference is considered minor.

Therefore, the change around 1998 is most likely attributable to urbanization and land-cover changes in the vicinity of the Suwon station. This explanation has been included in lines 391–393 of the revised manuscript.

[2] L.294-296: The vulnerability of urban areas to climate change involves a broader and more complex set of factors beyond this single physical characteristic (e.g., Dodman et al., 2022).

 Dodman, D., and coauthors, (2022): Cities, Settlements and Key Infrastructure. In: Climate Change 2022: Impacts, Adaptation and Vulnerability. Contribution of Working Group II to the Sixth Assessment Report of the Intergovernmental Panel on Climate Change. Cambridge University Press, Cambridge, UK and New York, NY, USA, pp. 907–1040, doi:10.1017/9781009325844.008.

We agree with the reviewer's comment and acknowledge that our previous wording may have led to misunderstanding. To avoid such ambiguity, we have revised the text to explicitly state that urban climate vulnerability is determined by multiple and broader factors, not limited to the effect of heat capacity. We have also cited relevant studies, including Dodman et al. (2022), to strengthen this clarification. The revision has been incorporated into lines 321–325 of the revised manuscript.

---

## Author Comment (AC2)

This study proposed a simplified mathematical model to explain the generation mechanism of urban heat islands (UHIs). This is a new attempt to understand the UHI phenomenon, however several limitations should be addressed.

We thank the reviewer for carefully reviewing and evaluating our manuscript. We also recognize that our approach has several limitations, and in the revision we have clarified the scope and intent of the study. We have addressed the specific points raised by the reviewer in the following responses, and we believe that these revisions have improved the clarity and completeness of the manuscript.

The main conclusion of this study is that the UHIs occur due to the large heat capacity of urban areas. This can be misleading. It is true that the urban surface materials usually have large heat capacity values. However, besides the material properties, the large thermal inertia in urban areas can be attributed to the inefficient longwave cooling and also the increase in surface area. Due to the presence of high-rise buildings, the reduced sky view factors in cities greatly reduce the longwave cooling at night. The incease in surface areas (envelops) due to buildings also contribute to storing more heat in urban areas compared to a flat surface. It would be great if the authors can quantify the individual contributions that result in large thermal inertia. If this is not possible, I strongly suggest that the authors clarify these points; otherwise, readers would simply think the UHIs are due to the large heat capacity of urban surface materials.

We fully acknowledge the reviewer's point that large thermal inertia in urban areas is not only caused by the heat capacity of urban surface materials but also by other factors such as inefficient longwave cooling and the increased surface area due to buildings. Since our study employs a simplified model, it has limitations in quantifying the individual contributions of these factors. To avoid any misunderstanding, we have revised the manuscript to explicitly state that the large heat capacity of urban materials should be considered as only one of the contributing mechanisms. These clarifications have been added in lines 24–26 and 430–432 of the revised manuscript.

I wonder if the dR/dT used in the defination of lambda (eq. 10) shows a clear diurnal variation or not in urban and rural areas. In the paper, constant values (11.8 or 30.9 W m-2 K-1) are used, but the use of a time-invariance value should be justified. If the dR/dT does vary with time, how can this temporal variation be considered in the theoretical model?

We understand the reviewer's comment. As noted, the variable used in this study can exhibit diurnal variation because it is influenced by the diurnal changes in several surface energy fluxes. However, such variations have little impact on the results at the primary timescale considered in our simplified model, and their overall order of magnitude remains similar. Therefore, we used the daily mean value as a representative input to reflect the typical characteristics of urban and rural areas. This revision has been added in lines 286–289 of the revised manuscript.

The authors argue that the stronger UHI intensity at night than at day is due to the larger heat capacity. This may be partially true. However, the authours seem not to consider the difference in boundary layer height in the daytime and in the nighttime. The much shallower boundary layer at night both in urban and rural areas amplifies the effects of differential heat fluxes. For example, the difference in sensible heat flux between urban and rural areas is larger in the daytime (let's say 200 W m-2) than in the nighttime (e.g., 20 W m-2). However,

the much deeper daytime boundary layer in the daytime (e.g., 1.5 km) than in the nighttime (e.g., 150 m) dilutes the differential heat flux effect. The authors are recommended considering this effect.

We understand the reviewer's concern. In our simplified model, the effect of boundary layer height was not explicitly considered. We agree that the shallower nighttime boundary layer amplifies the impact of differential heat fluxes, whereas the deeper daytime boundary layer tends to dilute them. To clarify this limitation, we have revised the manuscript to explicitly acknowledge the role of boundary layer height in modulating the diurnal variation of UHI intensity. These revisions have been added in lines 351–355 and 438–439 of the revised manuscript.

I am not sure if section 4.3 is necesseary for this study. I see a weak connection between section 4.3 and the previous sections. The authors would need to provide strong rationales or links of this section.

We acknowledge the reviewer's concern that Section 4.3 lacked a clear connection with the preceding sections. To strengthen the link, we have revised the Introduction (lines 96–99) to clarify the purpose of Section 4.3, stating that we utilize observational data in a megacity (Seoul) and a major city (Suwon) to compare the diurnal variation of the UHI intensity obtained from the theoretical model with that observed in reality, and to analyze long-term records to examine how UHI intensity evolves depending on the stage of urbanization.

Furthermore, we revised the beginning of Section 4.3 (lines 363–367) to explicitly build upon the previous subsection by emphasizing that the simplified day–night model qualitatively agrees with observations in Seoul and Yangpyeong, and that this agreement demonstrates the role of heat storage in strengthening nighttime UHI. Building on this insight, Section 4.3 then extends the analysis to long-term records, highlighting the contrasting UHI evolution in Seoul and Suwon with different urbanization rates.

We believe that these revisions provide a stronger rationale for Section 4.3 and clarify its connection to the earlier parts of the paper.

---

## Author Response (AR2)

General comments:

The manuscript has been improved by the authors' intensive effort to make thoughtful revisions in response to the previous comments. Through the revisions, the contribution of the proposed simple model in the broader context of urban climate research has become clearer. Sections 4.1 and 4.2 are also significantly improved by providing detailed descriptions and interpretations. Regarding Section 4.3, the interpretation seems to be correct, at least if we assume that the observed data were obtained under ideal conditions. On the other hand, there remains a question as to whether the contributions of factors other than the changes in heat capacity associated with urban development to the trends in $\Delta T$ and $\Delta DTR$ are relatively negligible. One of the reasons for that would be a lack of discussion about the absolute values of long-term changes in T and DTR at each of the urban and rural sites. Although I have the same opinion as reviewer 2's previous comment, that is "I am not sure if section 4.3 is necessary for this study", this should not affect the main parts and conclusions of this study. I think the proposed model is helpful in understanding the essence of the basic mechanisms of UHI.

We sincerely appreciate your positive evaluation of our revision efforts and your recognition of the improved clarity regarding the proposed model's contribution.

Regarding Section 4.3, we fully agree with your insight that the long-term trends in $\Delta T$ and $\Delta DTR$ observed in real urban environments are the result of complex interactions among various factors, not solely changes in heat capacity. Accordingly, we have revised the manuscript to explicitly acknowledge these limitations. We clarified that the purpose of Section 4.3 is not to claim that heat capacity is the exclusive driver, but to examine whether the physical behaviors and long-term trends observed are consistent with the mechanisms highlighted by our simplified time-integrated SEB framework.

Furthermore, as suggested, we have added a discussion on the absolute values of long-term changes in mean temperature and DTR at both urban and rural sites to provide a more comprehensive context. While we acknowledge, as you and Reviewer 2 noted, that Section 4.3 is not strictly necessary for the derivation of the study's main theoretical results, we have opted to retain it as a supplementary analysis that bridges our theoretical model with empirical observations. Finally, regarding Figures 8 and 9, we have increased the font size of the axis labels to improve readability.

Minor comments:

[1] l. 213 ηS --> η_S (S is a subscript)

Thank you for pointing this out. This has been corrected in line 213.

[2] l.263 "the difference in elevation between them is 38.4 (40.3)m" Which station is larger? Is $\Delta T$ used in the analysis applied to height correction?

Seoul is 38.4 m higher than Yangpyeong, whereas Suwon is 40.3 m lower than Icheon. To clarify this, a negative sign has been added before 40.3 in line 263.

The temperature differences ($\Delta T$) used in this study were not corrected for elevation. However, since the elevation differences between the paired stations remain nearly constant in time, their influence can be regarded as an approximately constant offset and thus does not affect the long-term trends of $\Delta T$ and $\Delta DTR$, which are the focus of this study.

[3] l. 284 : (colon) --> . (period)

Thank you for pointing this out. This has been corrected in line 284.

[4] l.291 and Eq. (5) Is S_0 (constant solar radiation) the value for a clear-sky day at local noon? If so and my understanding is correct, I wonder the solar radiation is overestimated by applying the noon value for the whole daytime (12 hours)?

$S_0$ is used as a representative forcing scale and is chosen to be of the order of the clear-sky solar radiation at local noon. We agree with the reviewer that applying a simplified daytime solar forcing can overestimate the daily integrated solar radiation compared to a realistic diurnal cycle. However, the primary objective of this theoretical model is not to derive precise quantitative values, but to elucidate the qualitative roles and fundamental mechanisms of key physical parameters.

Specifically, our aim is to analytically demonstrate how an increase in heat capacity leads to a damping of the diurnal temperature range, or how a decrease in albedo results in an increase in the mean temperature. In this context, while the exact temporal representation of solar forcing (e.g., noon value versus daily average) may affect the absolute magnitude of the results, it does not alter the core physical processes or the qualitative relationships that this study aims to highlight.

To clarify this point, an explanation of $S_0$ has been added in lines 291–292.

[5] l.377 The term "UCI" might be inappropriate here, because it is not clear whether the lower daytime temperature in urban than rural sites is due to the UCI or to other factors associated with location differences (e.g. elevation and latitude).

We agree with the reviewer that attributing the lower daytime urban–rural temperature difference observed at a specific time directly to a "UCI" can be influenced by site-specific factors such as elevation and latitude. However, the focus of this study is not on the absolute magnitude of the temperature difference at a given time, but rather on the long-term trend (slope) of the urban–rural temperature contrast.

The decrease in daytime UHI identified in this study represents a temporal change associated with the urbanization process over time. Such a change is expected to be only weakly related to factors such as elevation or latitude, which remain nearly constant in time. In this context, the weakening of daytime UHI is interpreted as a strengthening of UCI characteristics.

Nevertheless, to avoid potential ambiguity, the corresponding description has been revised with more cautious wording in lines 377-379.